# Biofeedback’s Effect on Orthosis Use: Insights from Continuous Six-Week Monitoring of Ankle Fracture Loading

**DOI:** 10.3390/s25030825

**Published:** 2025-01-30

**Authors:** Tobias Peter Merkle, Nina Hofmann, Christian Knop, Tomas Da Silva

**Affiliations:** Department of Trauma Surgery and Orthopaedics, Klinikum Stuttgart-Katharinenhospital, 70174 Stuttgart, Germanyc.knop@klinikum-stuttgart.de (C.K.); t.dasilva@klinikum-stuttgart.de (T.D.S.)

**Keywords:** ankle fracture rehabilitation, orthosis compliance, continuous monitoring, biofeedback

## Abstract

There is limited understanding of how well patients adhere to postoperative instructions following ankle surgery, particularly in outpatient settings regarding partial weight bearing (15–30 kg) and orthosis use. This study aims to assess orthosis compliance and load frequency over six weeks post-surgery using pressure-sensitive insoles, while also evaluating the effectiveness of continuous biofeedback. A total of 84 patients with isolated ankle fractures were enrolled. All participants were instructed to maintain partial weight bearing of 15–30 kg for six weeks with a lower leg orthosis equipped with insoles that continuously recorded daily step counts and maximum loads. In a prospective randomized design, the control group received no biofeedback, while the intervention group received audiovisual feedback whenever loads exceeded 20 kg. Adherence to the prescribed partial weight bearing and orthosis use was low in both groups, with only 10% of the control group wearing the orthosis by week three and overload occurring as early as week one. However, the implementation of biofeedback resulted in significant improvements in orthosis utilization (57.4% vs. 29.1%) and adherence to prescribed loading. The implementation of continuous biofeedback significantly enhanced adherence to prescribed loading and orthosis usage, highlighting its critical role in postoperative rehabilitation for ankle fractures.

## 1. Introduction

After surgery for an ankle fracture, it is a common practice to immobilize the leg using an orthosis. Additionally, patients are advised to bear only partial weight on the leg (Pfeifer, Grechenig) [1]. This therapeutic approach primarily aims to minimize the risk of secondary loss of reduction by providing stability and preventing complications caused by overloading or underloading. On one hand, there is concern about underloading, as potential risks include the development of osteopenia, muscle atrophy, joint stiffness, and deep vein thrombosis [2,3,4]. However, some load is desirable because weight bearing stimulates bone growth through osteoblastic activity [5], and bearing a partial weight of 20 kg results in nearly the same venous return as full weight bearing [6,7]. On the other hand, concerns about overloading arise from risks such as osteosynthesis failure, malunion of bone fragments, and delayed fracture healing due to excessive movement of fracture components [8]. The exact parameters of the target zone remain unclear to this day [9].

Given these potential risks associated with postoperative rehabilitation protocols, the need for precise methods to assess gait parameters and monitor rehabilitation progress becomes particularly evident. Despite established strategies, studies have shown significant challenges in practical implementation. A crucial issue in follow-up care is the long-term adherence to weight bearing instructions by patients. Regular training to maintain specific partial weight bearing levels using a discontinuous measurement method, such as a bathroom scale, also failed to improve compliance. Typically, patients receive guidance from a physical therapist on how to use crutches for partial weight bearing with the aid of a bathroom scale before being discharged from the hospital after surgery. However, patients are often unable to comply with these guidelines due to the lack of continuous feedback once they leave the scale behind [10,11]. According to Vasarhelyi et al., despite three days of practice on a scale, none of the 21 patients with a fracture of the lower extremity were able to comply with the prescribed partial weight bearing [8]. At the beginning of the study by Hereshko et al., the majority of the 18 patients exceeded the permitted weight bearing limit and did not improve even after 10 days of practice [12]. Hurkmans et al. reported that, in their study of 20 patients, despite daily training, the prescribed guidelines were not followed on the 7th postoperative day and further deteriorated by the 21st postoperative day [13].

These challenges highlight the necessity for robust monitoring mechanisms to ensure the effectiveness of rehabilitation in the home setting. Due to a lack of monitoring mechanisms, there have been only a limited number of studies to date that monitor prescribed rehabilitation protocols in a home environment for patients recovering from an ankle injury in their daily lives [8,14,15]. After discharge from the hospital, it is challenging to monitor outpatient patients during their daily activities to assess their gait, load bearing, and functional daily activities. It remains entirely unclear whether patients comply with these guidelines in everyday practice. Nevertheless, the expectation is for patients to adhere to the weight bearing limits we set, which are unlikely to be followed in their everyday lives. Additionally, this reliance extends to clinical studies without objective verification. Study protocols that rely on patient compliance to maintain specific loading levels should, therefore, implement continuous or repeated rapid monitoring of load bearing activities [14].

With the development of continuous measurement methods, monitoring of patients’ weight bearing in home and active functional environments has become possible. The use of a digital orthotic insole device is one of the recommended options, as it can promote compliance with rehabilitation protocols by providing audiovisual and haptic feedback [10,11,12,16,17,18]. Furthermore, this method is significantly superior to traditional teaching methods, such as the bathroom scale [16]. The introduction of digital technologies now offers new opportunities to make the rehabilitation of ankle fractures more effective and trackable. The continuous monitoring feature can determine not only how much the load recommendation is exceeded, but also how often and on which days. These devices can function as objective instruments for monitoring the healing process [10,14,17]. The utilization of ambulatory measurement devices not only permits the collection of objective data from patients outside of a gait laboratory, but also enables a significant improvement in compliance through the application of biofeedback. In earlier studies, it was demonstrated that, when coupled with the simultaneous use of live biofeedback, it is possible to guide patients’ loading into a predetermined target zone and significantly reduce the incidence of misloading [17,18]. This suggests that real-time biofeedback systems have the potential to enhance training and support research in patient populations.

Therefore, the aim of this study was to investigate patient compliance with our prescribed guidelines in the first six weeks after discharge to a home environment following surgical treatment of an ankle fracture. Additionally, we sought to assess whether the additional use of biofeedback would lead to measurable behavioral changes during this period.

## 2. Materials and Methods

### 2.1. Participants

Between November 2020 and January 2023, we enrolled a total of 84 patients with isolated ankle fractures who underwent surgical treatment at our hospital. Study participants had a minimum age of 18 years and were enrolled prospectively. Patients with multiple injuries, inadequate coordination or strength of the upper body, or gait disorders were excluded from the study. Participants who did not understand the study protocol were also excluded. All participants provided written informed consent. The ethics committee at the University of Tuebingen approved the study (protocol number 674/2021BO2). Two groups were formed through simple randomization: one intervention group received audiovisual feedback, while the control group did not receive any feedback (Figure 1). During the study, eight patients from the control group withdrew, citing concerns about data being shared with their health insurance, even though it was explicitly communicated beforehand that any sharing of data with health insurance was explicitly excluded.

### 2.2. Subject Characteristics

The study finally included 31 women and 35 men, with an average age of 45.2 years (SD 13.5), who met the inclusion and exclusion criteria. Their average weight was 78.3 kg (SD 13.5), and their average body mass index was 26.0 kg/m^2^ (SD 5.2). The dominant leg was 100% on the right side. All cases were low-energy trauma. The subject characteristics of both subgroups can be found in Table 1.

### 2.3. Experimental Protocol

All study participants were fitted postoperatively with a commercially available orthosis (SP Air Smart Walker, Sporlastic, Nürtingen, Germany) in their appropriate size. The measurement system employed in this study has been described previously in earlier studies [14,16,17,18]. This orthosis was equipped with an insole device (Sens2Go, Golex AG, Basel, Switzerland) that continuously measured the peak force of every step and recorded this data via a transmitter unit. For this study, feedback was activated for the intervention group whenever a load exceeding 20 kg was detected, while it was deactivated for the control group. The study participants were unable to make changes independently.

All patients received orientation in the three-point gait, both on even ground and on stairs, from our physiotherapists on the first postoperative day, in accordance with our in-house standards. Using a weighing scale, patients were instructed to apply weights within a target range of 15 to 30 kg. Prior to discharge, the intervention group received additional training from the authors on how to use the biofeedback system. They were instructed to adjust their weight with each step until the alarm signal was activated (Figure 2). They learned that longer intervals with the alarm signal, as well as instances where the alarm signal fails to activate, should be avoided. Discharge occurred after an average of 3.0 days (SD 1.5 days). All patients were informed that they were being continuously monitored. They were instructed to wear the orthosis for a total of 6 weeks, both day and night, and it could only be removed for personal hygiene and wound inspection. The patients were asked to keep a pain diary, recording their pain perception on a numerical pain scale (NRS) each week [19]. The peak pressure was continuously measured with each step and recorded during the clinical follow-up six weeks postoperatively. If no data were collected on a given day, that day was counted as if the orthosis had not been worn.

### 2.4. Statistical Protocol

With a coefficient of determination of R^2^ = 0.77, a statistical power of 0.8, and a significance level of α = 0.05, we obtained a sample size of *n* = 27. The normality of the data distribution (duration of orthosis wear and loads) was assessed using the Shapiro–Wilk test, which confirmed that the data were not normally distributed. Differences between the characteristics were compared with the Wilcoxon rank test. Data are reported as means and standard deviations. Influences of pain on weightbearing compliance were determined as Spearman’s rank correlation coefficient. JASP (Version 0.19.1, Department of Psychological Methods, University of Amsterdam, 59 Amsterdam, The Netherlands) was used for statistical analysis.

## 3. Results

### 3.1. Duration of Orthosis Wear

The results shown in Figure 3 demonstrate a clear and statistically significant difference in participant retention between the intervention group and the control group. In the overall cohort over the entire 6-week period, the average duration of orthosis use in the intervention group was 57.4%, while in the control group, it was 29.1%, demonstrating a statistically significant difference (*p* < 0.001). The analysis revealed that the intervention group exhibited significantly higher retention compared to the control group during weeks 2 to 6. The mean values of retention declined in both groups over the six weeks. This decline was more pronounced in the control group than in the intervention group.

### 3.2. Loads and Number of Steps

In Figure 4, the number of steps is divided into three load zones—underload (<15 kg), target zone (15–30 kg), and overload (>30 kg)—over time.

The control group consistently had more steps in the overload zone from the beginning compared to the intervention group. This overload continued to increase from week one to week two. Looking at the percentage share (Figure 5), it increased from 27% to 45%, while the steps in the target zone decreased. The data from the control group starting from the third postoperative week are no longer valid and are displayed in gray, as the number of loading data points was too low due to insufficient willingness to wear the orthosis.

The intervention group managed to maintain approximately two-thirds of their steps in the target zone throughout the entire period. Only in the last week did every second step fall into this zone (Figure 5). A gradual increase in the average number of steps per week was observed in this group (week 1 = 2627, week 2 = 5692, week 3 = 4817, week 4 = 6104, week 5 = 6729, week 6 = 7828) (Figure 4). This is also evident in the control group during the first two weeks, with an average of 4321 steps in the first week and 6581 steps in the second week.

### 3.3. Pain

The analyses indicate that there was no statistically significant difference in pain perception between the intervention group (with biofeedback) and the control group in any of the six weeks (Figure 6). The standard deviations are relatively high across all groups and weeks, suggesting considerable variability in the participants’ pain perception (Figure 7).

In both groups (control and intervention), it was noted that there was no significant correlation between the duration of orthosis wear and pain sensations (Table 2). There was a very weak negative correlation in the control group during week 1 and a weak positive correlation in week 2. This may suggest a slight tendency for higher loads to be associated with increased pain. However, the *p*-value in both weeks was greater than 0.05, indicating that this correlation was statistically non-significant. In the intervention group, no significant correlation between load and pain sensations was observed in either week. All *p*-values were above 0.05, suggesting that the correlations were statistically non-significant, and thus, no strong relationship between the variables was present.

No cases of implant failure were observed. There were four cases of wound healing disturbances in both groups, which healed without further surgical intervention over time.

## 4. Discussion

This study investigated patient adherence during the first six weeks of rehabilitation after ankle osteosynthesis with the use of a loading insole. It utilized a typical postoperative rehabilitation protocol commonly used in clinical practice, which involves partial weight-bearing within a target zone between 15 and 30 kg in a below-knee orthosis for a duration of six weeks. The results indicate that patients began to disregard their surgeon’s instructions immediately after being discharged from the hospital.

### 4.1. Duration of Orthosis Wear

The analysis of orthosis wear duration reveals that both the control group and the intervention group experienced a decline in usage over the six weeks. Averages decreased across both groups during this period, with a particularly pronounced decline in the control group from weeks 2 to 6. During this time, the frequency of wear fell drastically from week 2, averaging just above 10% between weeks 3 and 5 and dropping to a single-digit percentage in week 6. In contrast, the intervention group demonstrated significantly better adherence starting in week 2. The orthosis was worn in this group for an average of over 50% of the time during weeks 2 to 5, before usage declined to about 40% in week 6. Although pain perception decreased over time in both groups, no significant difference between the groups was evident. The correlation analysis further revealed that pain was not a significant factor in the decision to remove the orthosis in this sample. This suggests that other factors, such as practical challenges, a lack of understanding of the orthosis’s benefits, or social and professional obligations, may have played a larger role. One aspect to consider is the Hawthorne effect, which states that individuals change their behavior when they know they are being observed. This could explain why some patients primarily wore the orthosis because they were aware that their data were being monitored. It is possible that their willingness to wear the orthosis would have been even lower without continuous oversight. This may also clarify why eight patients chose to withdraw from the study, possibly out of concern that their actual use of the orthosis would be viewed negatively.

Braun et al. already expressed concerns in their study regarding whether the insole was used outside of the hospital, as they noted early returns and had no records after day five in some cases. Additionally, the concerns raised by this research group that previous studies without continuous measurements should be viewed with particular caution are echoed in our study [14]. The results illustrate that patients did not consistently adhere to their prescribed guidelines, despite the fact that the use of biofeedback improved compliance. However, further measures are necessary for long-term adherence improvement. We see significant potential in the further development of the software for these technological aids, such as exercise and motivation programs that can encourage patients to wear the orthosis as recommended through positive reinforcement. We are familiar with similar motivational approaches from sports watches and fitness trackers [20]. Providing insight into the stored device data and implementing a live program that reflects therapy progress could support weekly therapy sessions and be used to set new, achievable goals. Additionally, regular follow-up appointments could help to identify issues early and make necessary adjustments. Rigid rehabilitation protocols should be reconsidered, as traditional postoperative informational consultations, like those we conducted, proved insufficient to ensure the success of therapy.

### 4.2. Loads

The analysis of load showed that the control group had a higher mean number of loads during the first two weeks compared to the intervention group, even though no statistically significant difference was found. This suggests that the group without biofeedback was more mobile during this period. We had already demonstrated that the additional feedback from biofeedback encouraged participants to act more cautiously. Participants adjusted their spatiotemporal parameters: walking speed decreased, step duration increased, and step length and step frequency decreased simultaneously [18].

The results of this study further indicate that compliance with the prescribed load decreased over time. In particular, the number of steps in the overload zone increased throughout the weeks. Previous studies have documented similar trends, confirming that adherence to load recommendations decreases after discharge into the home environment. Chiodo et al. examined patient compliance with various foot injuries in their home environments by using pressure-sensitive film in the cast. This study showed a non-compliance rate of 27.5% regarding weight-bearing recommendations [15]. Braun et al. demonstrated that less than half of the patients remained compliant with the load recommendations after 14 days, and compliance further declined thereafter [14]. Hurkmans et al. were able to show that audio feedback improved the accuracy of partial weight bearing during training; however, the desired accuracy during unsupervised walking could not be maintained in the hospital by the 7th day and continued to be ineffective at home by the 21st day [13].

Despite the low compliance rate regarding the load recommendations, our study observed no implant failures or delayed fracture healing during the 6-week X-ray follow-up, and the number of wound healing complications was equal in both groups (four cases each, at the lateral malleolus, without the need for surgical revision). This suggests that patients may be able to adjust their follow-up care independently without negatively impacting their healing outcomes. However, the aim of this study was to evaluate the accuracy of the rehabilitation recommendations rather than their clinical effects. Our results underscore the effectiveness of rehabilitation protocols that incorporate continuous measurement methods with biofeedback. Such protocols significantly improve and prolong adherence compared to conventional training methods, such as the use of a personal scale, as we had recently demonstrated in a randomized controlled study [16]. The adherence to the loads within the target zone in our study highlights the necessity of using a continuously measuring gait analysis tool in the early rehabilitation phase. The continuous monitoring feature of our system allows the determination of not only how much the weight recommendations are exceeded, but also how frequently and on which days. The system used in our study features a memory function that enables easy retrieval of data with a mobile device, allowing a therapist to provide individualized treatment to the patient after reviewing the recorded load data.

The perception of pain did not play a significant role in our patient cohort. The results indicate that there was no significant correlation between wear duration or load and pain in the groups, suggesting that other factors, such as motivation or psychosocial aspects, may have played a larger role. The collection of other parameters was not feasible due to data protection regulations concerning mobile devices, such as the number of physical therapy sessions or manual lymphatic drainage. Warmerdam et al. described significant issues regarding patients’ willingness to use their devices in everyday life. Patients found that their insole devices were cumbersome, requiring daily data storage and battery changes, which many considered excessive following their injury. Only 6 out of 16 patients were able to collect sufficient data over the 6-week period that could be used for research purposes, as the remaining patients did not consistently wear the insoles [4]. We were also unable to perform valid statistical evaluations in the control group during weeks 3–6 due to the small sample size, which is why these data are presented in gray in the figures and tables.

The concept of rigidly prescribed partial loading combined with immobilization using either an orthosis or a cast following an ankle fracture has been the subject of critical evaluation for several years. Baumbach et al. pointed out in their systematic review that pain-dependent weight bearing did not appear to increase complication rates, raising the question of whether patients were already able to exercise correctly without any load instructions [11]. It is not known whether this method promotes fracture healing or reduces complications, as the researchers have not yet had objective means to continuously measure or investigate the load on the extremities and its relationship to fracture healing [21]. Due to the often-high heterogeneity of study participants and fracture types in these studies, as well as a low patient count, a consensus recommendation has not yet been established. Studies with high levels of incidence and homogeneous study populations are lacking [2]. In our opinion, not every patient should be treated uniformly. At present, there are too many unresolved variables, such as age, body weight, and fracture type. The identification of these variables should be the subject of future studies. Without objective data collection, surgeons have to continue relying on existing post-treatment concepts in the near future.

### 4.3. Limitations

The present study has limitations that affect the generalizability of the results and their clinical relevance. The findings primarily pertain to the specific population of study participants, and it remains unclear whether they can be applied to other groups with differing demographic characteristics or pre-existing conditions. A larger and more diverse sample could enhance generalizability. Another limiting aspect is the measurement method used, which cannot differentiate whether patients are wearing the orthosis at night or intermittently; this may influence the interpretability of the results. Furthermore, statistical significance does not necessarily equate to clinical relevance, which necessitates a more detailed evaluation of the practical implications of the differences in cast-wearing duration on the healing process and functional outcomes of the patients. The study design provided only a snapshot of the early rehabilitation phase and did not yield meaningful information regarding the subsequent rehabilitation process. Additional analyses examining the impact of cast-wearing duration on patient outcomes would be beneficial. However, this was not the goal of the study, which aimed to investigate compliance with the prescribed guidelines during the first six weeks post-discharge. It is also noteworthy that the study focuses on a specific form of audiovisual biofeedback, which was implemented at a particular load threshold of 20 kg, and the transferability of the results to other forms of biofeedback remains unclear. Finally, Warmerdam et al. found that significant costs were incurred due to technical problems with their devices [4]. Similarly, in our study, no health insurance provider was willing to cover the costs of the devices, even though they were in the low three-digit range and did not present any significant technical issues. However, the study demonstrates that the monitoring of weight measurement using the presented insole is feasible, which could be beneficial in the development of individualized rehabilitation protocols, especially for patients with low mechanical stability or for older patients.

## 5. Conclusions

The compliance with adherence to a prescribed rehabilitation protocol without feedback was low in our study cohort. Previous studies conducted without continuous measurements should, therefore, be viewed with particular caution. The biofeedback provided by the insole was able to contribute to a significant improvement in compliance, although the motivation to utilize this device still needs to be enhanced. Additional influencing factors can now be objectively investigated through intervention studies to determine the therapeutic value of these factors and to define new, individualized rehabilitation standards.

## Figures and Tables

**Figure 1 sensors-25-00825-f001:**
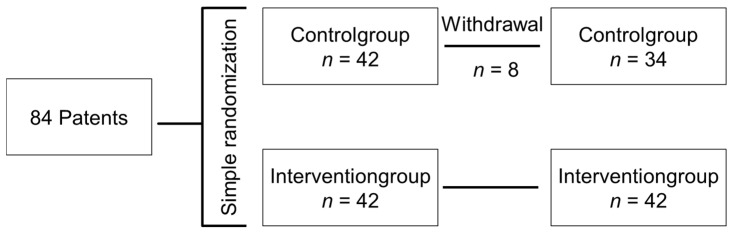
Flowchart of the study. The study ultimately included 34 patients in the control group and 42 patients in the intervention group.

**Figure 2 sensors-25-00825-f002:**
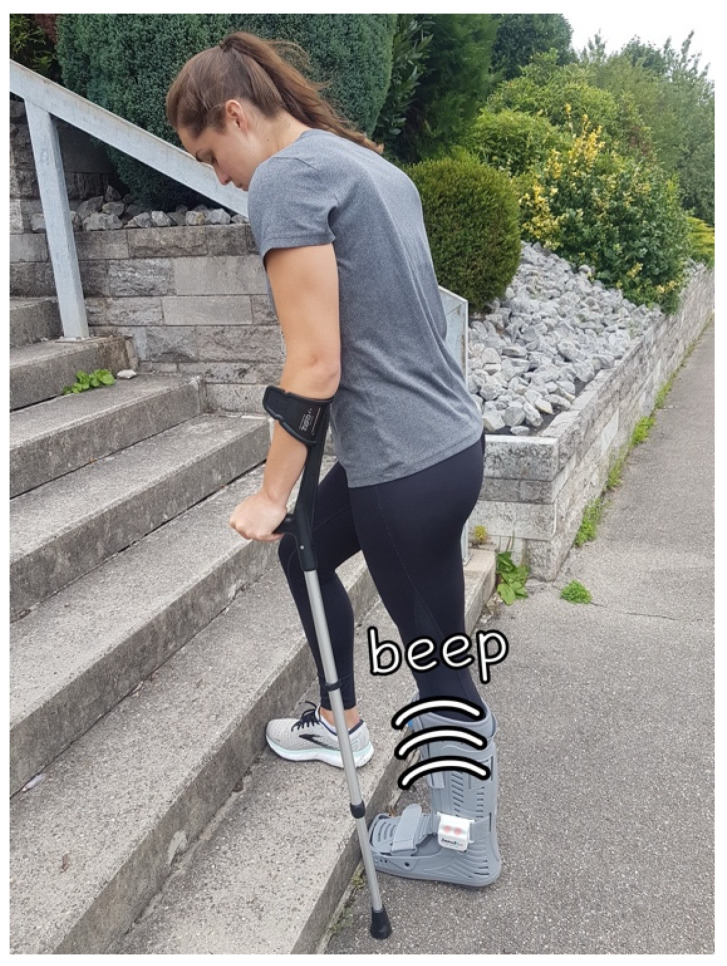
A patient walking up a staircase using underarm crutches with the applied orthosis. The orthosis emits a clearly audible warning signal in addition to a red light.

**Figure 3 sensors-25-00825-f003:**
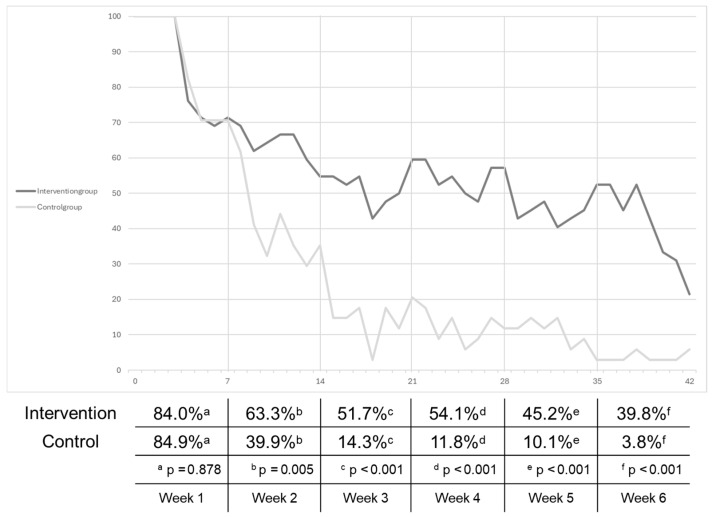
Descriptive statistics of participant retention for the intervention and control groups, stratified by week. The accompanying table shows the percentage of orthoses worn per week, divided between the two groups. Paired comparisons between the intervention and control groups were conducted for each of the six weeks. The Wilcoxon signed-rank test was employed, as normal distribution was not assumed.

**Figure 4 sensors-25-00825-f004:**
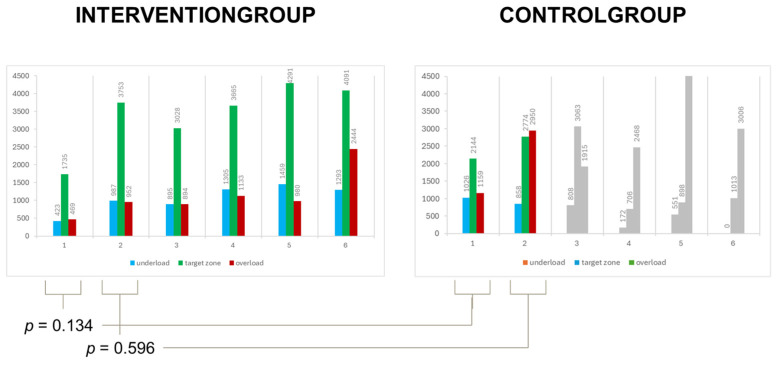
The loads for both groups per week. On the left side, we see the intervention group, and on the right side, the control group. Weeks 3–6 are shaded out in the control group, as no valid conclusions could be drawn due to the low wear duration in this group.

**Figure 5 sensors-25-00825-f005:**
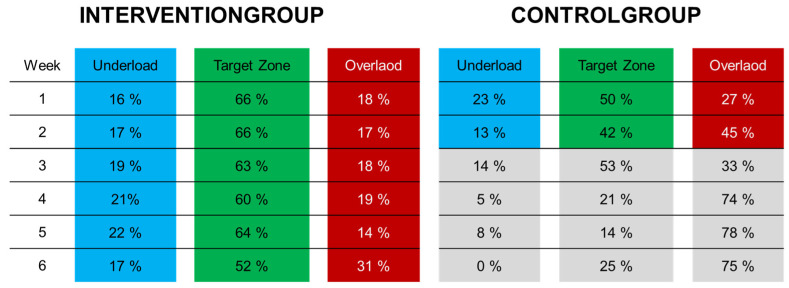
The percentage share of steps in the three load zones in both groups. Again, we have grayed out weeks 3–6 in the control group, as there were too few measurements to make a valid statement.

**Figure 6 sensors-25-00825-f006:**
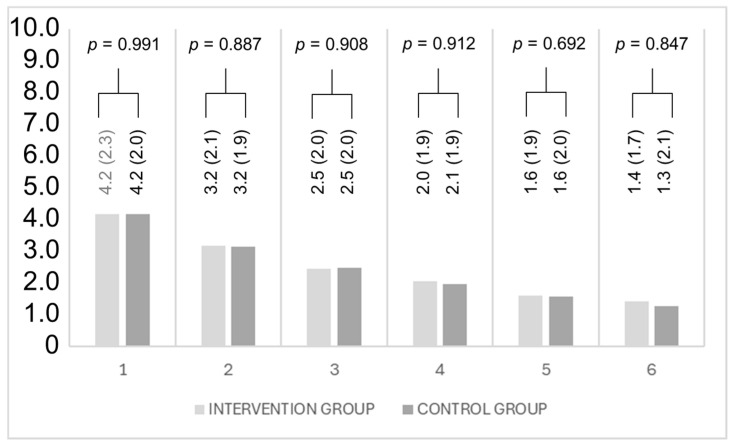
This figure presents descriptive statistics of pain perception for the intervention and control groups, separated by week. Postoperative pain levels were measured using the numeric rating scale (NRS). The data are presented as means and standard deviations. The Wilcoxon signed-rank test was used, as normal distribution of the data was not assumed.

**Figure 7 sensors-25-00825-f007:**
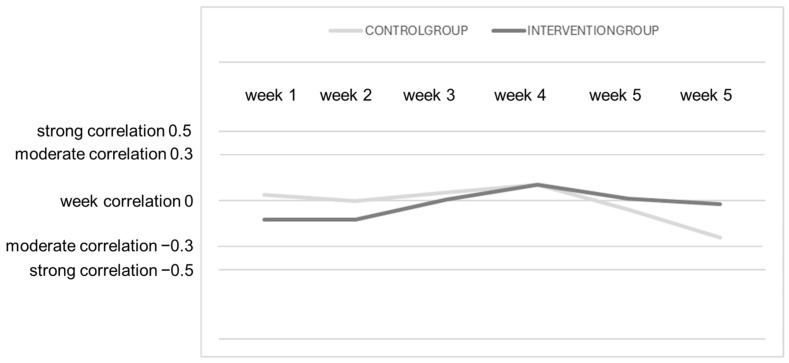
The results of the Spearman’s rank correlation test used to determine correlations between pain sensations and the duration of orthosis wear.

**Table 1 sensors-25-00825-t001:** The subject characteristics of the two subgroups: those without biofeedback (control group) and those with biofeedback (intervention group). Age (*p* = 0.365), weight (*p* = 0.135), and BMI (*p* = 0.343) were not significantly different.

Characteristic	Intervention-Group	Control-Group
**No. of Patients**	42	34
**Sex** **Female** **Male**	22 20	19 15
**Mean age**	43.9 (SD 13.9)	46.9 (SD 13.0)
*Intervention group (percentile)*	25th 31.3, 50th 46.0, 75th 54.0	
*Control group (percentile)*	25th 37.5, 50th 47.0, 75th 58.3	
**Mean Weight**	75.8 (SD 13.5)	80.3 (SD 13.2)
*Intervention group (percentile)*	25th 71.0, 50th 80.0, 75th 86.0	
*Control group (percentile)*	25th 68.0, 50th 73.0, 75th 80.0	
**BMI**	26.5 (SD 4.0)	25.3 (SD 6.3)
*Intervention group (percentile)*	25th 23.9, 50th 25.3, 75th 28.1	
*Control group (percentile)*	25th 23.5, 50th 24.8, 75th 26.0	
**Injured Side** **left** **right**	30 12	15 19
**Fracture Type** **Isolated** *, of these classified as Weber * *B/Weber C* **bi/trimalleolar** *, of these classified as * *Weber B/Weber C*	21 *18/3* 21 *11/10*	19 *11/8* 15 *6/8*

**Table 2 sensors-25-00825-t002:** The exact rho values, along with the corresponding *p*-values in parentheses.

	Week 1	Week 2	Week 3	Week 4	Week 5	Week 6
INTERVENTION	−0.136 (0.39)	−0.136 (0.804)	0.01 (0.948)	0.115 (0.467)	0.013 (0.933)	−0.023 (0.886)
CONTROL	0.044 (0.803)	−0.003 (0.987)	0.056 (0.753)	0.112 (0.530)	−0.065 (0.714)	−0.269 (0.124)

## Data Availability

We are willing to make the raw research data available for further exchange. Requests can be made directly to the corresponding author.

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
