# Peer review of "Biofeedback’s Effect on Orthosis Use: Insights from Continuous Six-Week Monitoring of Ankle Fracture Loading"

_sensors, 2025, doi:10.3390/s25030825_

Round 1
Reviewer 1 Report
Comments and Suggestions for Authors
The topic of the article has clinical interest due to the lack of reliable tools to ensure adherence and compliance of patients who need to use an orthosis after surgical treatment, once they are in their home environment. The introduction sufficiently provides the context and justification for the research problem posed. However, there are some important considerations that would clarify the follow-up in the development of the experimental work, as established by the CONSORT statement for reporting experimental clinical studies. Specifically, the flow diagram that clarifies the different phases and the subjects who were recruited, assigned, and analyzed in each of them. In Table 1, it is important to include an analysis of the baseline differences in the pre-treatment variables, as the quartiles of the distribution of the values are not provided.
A sample size calculation has not been included.
In lines 137 and 138, it would be necessary to include the reference for the scale used to assess the self-perceived degree of pain. I understand it is a VAS?.
The difference in retention in the control group is very striking. It is noteworthy that despite the premature removal of the orthosis by the control group, there are no differences in the perceived degree of pain, nor are there differences in other clinical circumstances. It is also unclear whether there were differences in the time to restore painless walking, overload complications, etc. This could serve as a starting point for researchers to review such protocols, especially in the long term, and to consider professional follow-up by physiotherapists once the clinical center period is over. On the other hand, I would have liked to read in the discussion or as a possible limitation whether the determination of the load based on specific kg, instead of a proportion relative to the patient's specific weight, could have been a limiting factor (both excessive and insufficient) in the challenge or effort that each participant must individually exert. I agree that the effect of being observed during the study has been an importnat factor in this study.
Reviewer 2 Report
Comments and Suggestions for Authors
Excellent article that is thorough and the narrative flows nicely as it is well-organized. The discussion raises excellent points regarding compliance and the fashion in which feedback influence patient behavior. This seems to be a superb effort that could easily lead to additional research phases (not within the current scope). Just a few thoughts to consider, some quite minor but hopefully helpful.
Lines 143-145: Please expand on which specific data was tested with the Shapiro-Wilcox test and found to be not normal. It is unclear if that this was expected and why – trivial, yes, but caused the reader to pause.
Figure 2: Excellent characterization of the results. Please consider proving some clarity that the percentages provided under the curve related to each week; yes, this is inferred from the narrative description but providing a label for each week (on the graph) drives home an important aspect of your research outcomes.
Figure 3 is also commendable, though is micro-coded; Lines 173+ describe what is needed, but once again the unique dataset and its impact seem are diminished with the Figure as-is.
Lines 237-244 offer significant insight and underscore the issues this research effort identifies or underscores. Compliance is an issue that did not seem to go away with even this study. It was hoped possible “lessons learned” on feedback speculative ideas on patient engagement for improved compliance could have been included; this does not take away from the manuscript in current form.
Round 2
Reviewer 1 Report
Comments and Suggestions for Authors
I want to thank the author to the efforts to improved the final version of the manuscript.